# Prolonged SARS-CoV-2 T Cell Responses in a Vaccinated COVID-19-Naive Population

**DOI:** 10.3390/vaccines12030270

**Published:** 2024-03-04

**Authors:** Vassiliki C. Pitiriga, Myrto Papamentzelopoulou, Kanella E. Konstantinakou, Irene V. Vasileiou, Alexandros D. Konstantinidis, Natalia I. Spyrou, Athanasios Tsakris

**Affiliations:** 1Department of Microbiology, Medical School, National and Kapodistrian University of Athens, 75 Mikras Asias Street, 11527 Athens, Greece; atsakris@med.uoa.gr; 2Molecular Biology Unit, 1st Department of Obstetrics and Gynecology, National and Kapodistrian University of Athens, 11528 Athens, Greece; mpntua@yahoo.gr; 3Bioiatriki Healthcare Group, Kifisias 132 and Papada Street, 11526 Athens, Greece; nkonstantinakou@bioiatriki.gr (K.E.K.); irenebasileiou@gmail.com (I.V.V.); konstantinidisalex92@gmail.com (A.D.K.); nataspyrou@gmail.com (N.I.S.)

**Keywords:** cellular immunity, COVID-19, SARS-CoV-2, T cell immunity, vaccines, vaccination

## Abstract

Introduction: Exploring T cell response duration is pivotal for understanding immune protection evolution in natural SARS-CoV-2 infections. The objective of the present study was to analyze the T cell immune response over time in individuals who were both vaccinated and COVID-19-naive and had undetectable levels of SARS-CoV-2 IgG antibodies at the time of testing. Methods: We performed a retrospective descriptive analysis using data extracted from the electronic medical records of consecutive adult individuals who underwent COVID-19 immunity screening at a private healthcare center from September 2021 to September 2022. The study participants were divided into three groups according to the post-vaccination time period, as follows: group A (up to 3 months), group B (3–6 months), and group C (>6 months). T cell response was evaluated using the IGRA methodology T-SPOT^®^.COVID. Results: Of the total number of subjects (n = 165), 60/165 (36.4%) had been vaccinated in the last 3 months (group A), 57/165 (34.5%) between 3 and 6 months (group B), and 48/165 (29.1%) at least 6 months prior to the examination day (group C). T cell positivity was reported in 33/60 (55.0%) of group A, 45/57 (78.9%) of group B, and 36/48 (75%) of group C (*p* < 0.007). No statistically significant differences were revealed in the spot-forming cell (SFC) count among groups, with mean SFC counts of 75.96 for group A, 89.92 for group B, and 83.58 for group C (Kruskal–Wallis test, *p* = 0.278). Conclusions: Our findings suggest that cellular immunity following SARS-CoV-2 vaccination may endure for at least six months, even in the presence of declining or absent IgG antibody levels.

## 1. Introduction

Vaccination against severe acute respiratory syndrome coronavirus 2 (SARS-CoV-2) has significantly contributed in protecting individuals from the severe coronavirus disease of 2019 (COVID-19) and mortality, since both humoral and T cell responses are elicited [1]. Maximum levels of immune responses generated by any of the European Medicines Agency (EMA)-approved vaccines are detected no earlier than 14 days after completing the vaccination schedule [2]. Post-vaccination-generated immunity levels in healthy individuals determine the capacity of vaccines to prevent initial infection and control the progression of the infection and transmission of SARS-CoV-2. In particular, the investigation of T cell immune response longevity facilitates the understanding of how immune protection develops and persists during the SARS-CoV-2 infection but also provides useful information for the evaluation of current vaccines in a vaccinated SARS-CoV-2-naive population [3,4].

T cell immunity following a complete two-dose vaccination is less well studied than humoral responses, since its importance had been underestimated at least at the beginning of the COVID-19 pandemic, where T cell responses were difficult to study [5]. SARS-CoV-2 vaccines have demonstrated a remarkable efficacy, but the duration and nature of T cell responses to vaccination is of great research interest. In an early longitudinal study, researchers showed that mRNA vaccines activated antigen-specific CD4^+^ T cells and, after a booster dose, T cell responses in SARS-CoV-2-naive individuals were improved [6]. A subsequent multicentre prospective study demonstrated that the vaccines activated T cells, particularly CD4^+^ T cells, contributing to durable protective immunity [7]. In a large longitudinal study, CD4^+^ T cells, CD8^+^ T cells, and memory B cells were detected in healthy volunteers almost six months after the vaccination [8]. Moreover, enhanced cellular responses with lower humoral immunity in SARS-CoV-2-naive individuals were detected up to seven months post complete vaccination [9].

On the other hand, there are studies demonstrating that cellular immunity co-exists with humoral responses for a longer period; however, a gradual waning over time is observed for both immune responses. In detail, both humoral and cellular SARS-CoV-2-specific responses were present at least up to two months post-vaccination in most SARS-CoV-2-unexposed individuals that received two inactivated COVID-19 vaccine doses [10]. Another study revealed that upon mRNA vaccination, both the humoral and cellular responses of SARS-CoV-2-naive nursing home residents significantly waned over the course of six months, with an estimated half-life of the antibody response of 47 days [11]. Similarly, a prospective cohort study revealed a decline in both humoral and cellular responses within six months after a two-dose vaccination [12]. As previously published, substantial declines in neutralizing antibody titers six months post-vaccination were reported, while memory CD4^+^ and CD8^+^ T cells, as well as memory B cells exhibited small reductions [13]. Waning antibody responses were also observed between 3 and 6 months after vaccination in uninfected individuals with a milder decrease in spike-specific T cell responses [14]. 

Based on our current understanding and building upon our recently published study on SARS-CoV-2 cellular immunity [15], we undertook the current retrospective study with the primary focus of examining T cell immunity levels. This investigation utilized enzyme-linked immunosorbent T-SPOT assays (ELISpot) and involved individuals who had been vaccinated, were SARS-CoV-2-naive (non-exposed), and exhibited an absence of antibody responses during testing.

## 2. Methods

### 2.1. Study Design

A descriptive retrospective analysis was conducted using data obtained from the electronic medical records of consecutive adult individuals who underwent COVID-19 T cell immunity testing at the “BIOIATRIKI” Healthcare Center. This medical facility, located in the Attica region, implemented the testing after individuals’ testing order, as part of a SARS-CoV-2 immunity screening effort throughout the pandemic. The study population consisted of individuals who were COVID-19-naïve and had been fully vaccinated but tested negative for SARS-CoV-2 IgG antibodies on the day of testing. To be included in the study, participants had to meet specific criteria, including the absence of COVID-19 symptoms, no known contact with a COVID-19 patient, no prior diagnosis of SARS-CoV-2 infection based on various test results (molecular, immunochromatography, and antibody tests), completion of the initial vaccination series, and no receipt of booster doses up to the day of testing. The date of the initial vaccination completion was designated as the vaccination date. In cases where two doses were required for complete vaccination, the date of the second dose administration was designated as the vaccination date. Immunocompromised patients were not included in the study. We categorized individuals as immunocompromised if (a) they had primary severe immunodeficiencies present from birth and (b) they became immunocompromised due to medical treatments, including transplantation, specific autoimmune diseases affecting the immune system, and cancer patients regularly using drugs intended to suppress the immune system or inadvertently causing immune suppression (e.g., chemotherapy and oral steroids). Additional medical history data were collected using structured questionnaires regularly filled out by all participants during the testing process. Although detailed information about the specific vaccines administered was not available for all participants, the mRNA vaccines were administered to the vast majority of the Greek population during the study period. The analysis of T cell responses was distributed into three time periods after vaccination, as follows: group A (up to 3 months), group B (3–6 months), and group C (>6 months). The study was approved by the institutional review board during the sixth annual meeting held on 29 June 2021. 

### 2.2. Laboratory Methods

#### 2.2.1. Enzyme-Linked Immunosorbent T-SPOT (ELISpot) Assay for IFN-g T Cell Response Detection

To assess the T cell response to SARS-CoV-2 infection, the IGRA methodology T-SPOT^®^.COVID (Oxford Immunotec, Oxfordshire, UK) was employed. This standardized ELISpot-based technique detects the T cell immune response to SARS-CoV-2 in whole blood, utilizing the T-SPOT Technology with an antigen mix based on SARS-CoV-2 structural proteins, spike (S), and nucleocapsid (N). Blood samples were collected in lithium heparin tubes and the T cell Xtend reagent (Oxford Immunotec) was added. Peripheral blood mononuclear cells (PBMCs) were isolated, washed, and counted before being introduced into the test. Approximately 250,000 cells/well were plated into four wells of a 96-well plate. Two antigen peptide pools were added to specific wells, with the T cell mitogen phytohemagglutinin in the positive control well and cell culture media alone in the negative control well. Following 16–20 h of incubation, the wells were washed, and a conjugated secondary antibody capable of binding to any captured IFN-γ on the membrane was added. Unbound IFN-γ was removed through washing and a substrate was added to produce characteristic dark spots, indicating areas of IFN-γ presence. Spot-forming cells (SFCs) were manually counted by expert technologists. Results for N and S antigens were reported separately. Results were considered ‘invalid’ if the negative control had more than 10 SFCs or the positive control had fewer than 20 SFCs when the antigen wells were non-reactive. The test cutoff was predefined at 6 SFCs for each antigen, with a borderline zone of +/−1 SFCs introduced to account for the potential test variability around the cutoff. Consequently, results were reported as ‘S and/or N reactive’ when SFCs in the S and/or N antigen wells minus the negative control were ≥8, ‘S and N non-reactive’ when SFCs in the respective antigen wells minus the negative control were ≤4, and ‘S and/or N borderline’ when SFCs in the respective antigen well minus the negative control were 5, 6, or 7.

#### 2.2.2. SARS-CoV-2 IgG Antibodies

Levels of SARS-CoV-2 IgG antibodies were quantified in participants’ blood sera on the same day using the SARS-CoV-2 IgG II Quant assay from Abbott Laboratories, employing a quantitative method. This is an automated, two-step chemiluminescent microparticle immunoassay (CMIA) used for the qualitative and quantitative determination of IgG antibodies to the receptor binding domain (RBD) of the S1 subunit of the spike protein of SARS-CoV-2, in human serum and plasma on the Alinity i system. The RBD sequence used was derived from the WH-Human 1 coronavirus, GenBank accession number MN908947. The manufacturer’s specified analytical measurement interval is 21–40,000 arbitrary units (AU)/mL, with a reported positivity cutoff of ≥50 AU/mL.

### 2.3. Statistical Analysis

In terms of statistical analysis, the chi-square (Χ^2^) test was applied to assess the differences between categorical variables, and either a one-way ANOVA or a T test was used to compare continuous variables. For non-parametric continuous variables, the Kruskal–Wallis test was executed. Spearman’s rank test was employed to investigate the potential correlations between T-SPOT SFC counts and the time interval post-vaccination. The kinetics of SFC counts over time were illustrated using plotting charts depicting the curve estimation for linear regression models. All statistical analyses and graphical demonstrations were performed using IBM SPSS Statistics version 28 (SPSS). Results were considered statistically significant when the *p*-value was less than 0.05.

## 3. Results

### 3.1. Participants’ Demographic Data and Comorbidities

From all individuals tested by our medical center for cellular immunity screening, 165 were included the study; 91 females (55.2%) and 74 (44.8%) males aged from 17 to 92 years old (mean 58.8 ± 15.8 years). The study participants’ demographic characteristics and medical information are presented in Table 1. All individuals had antibody levels below the lower assay limit of 50 AU/mL.

Among them, 60/165 (36.4%) had been vaccinated within the last 3 months from the examination day (group A), 57/165 (34.5%) had been vaccinated 3–6 months before the examination day (group B), and 48/165 (29.1%) had been vaccinated at least 6 months before the examination day (group C).

In group A, the sex percentage was 36/60 females (60%) and 24/60 males (40%), in group B 30/57 females (52.6%) and 27/57 males (47.4%), and in group C 25/48 females (52.1%) and 23/48 males (47.9%) (chi-square test, *p* = 0.64). 

The groups did not exhibit any differences in age. More specifically, the mean age was 58.0 ± 15.2 years in group A, 59.1 ± 17.5 years in group B, and 59.5 ± 14.5 in group C (one-way ANOVA test, F = 0.14, *p* < 0.86). The T-SPOT measurements of T cell responses to the S antigen at the respective ages of the total participants are presented in Figure 1.

### 3.2. T Cell Positivity Rate among Groups and in Total Participants

Of the total number of participants, a positive T cell response was produced in 114/165 (69.1%). Borderline measurements were reported in 4/165 (2.4%) and non-reactive measurements in 47/165 (28.5%). 

Among groups, the proportions of borderline and non-reactive results were as follows: In group A, a positive S response was detected in 33/60 (55%), borderline in 1/60 (1.7%), and non-reactive in 26/60 (43.3%). In group B, a positive S response was detected in 45/57 (78.9%), borderline in 3/57 (5.3%), and non-reactive in 9/57 (15.8%). In group C, a positive S response was detected in 36/48 (75%), borderline in 0/48 (0%), and non-reactive in 12/48 (25%).

A statistically significantly higher proportion of T cell positivity rate was observed in groups B and C compared to group A. More specifically, among the three groups, T cell reaction against the S antigen was reported in 33/60 (55.0%) of group A, 45/57 (78.9%) of group B, and 36/48 (75%) of group C (chi-square, *p* < 0.007). T-SPOT results for the N antigen were negative (SFC ≤ 4) for all participants.

### 3.3. Quantitative IFN-g Response against SARS-CoV-2 S Antigen

The median SFC count for the S antigen was 11.50 (ranging from 0 to 275) in group A, 15 (ranging from 0 to 110) in group B, and 14 (ranging from 0 to 72) in group C. The Kruskal–Wallis test revealed no statistically significant differences in the SFC count among the various groups, χ^2^ (2) = 2.564, *p* = 0.278, with mean rank SFC counts of 75.96 for group A, 89.92 for group B, and 83.58 for group C.

### 3.4. T Cell Response Based on the Days Following Vaccination

The mean time period after vaccination for all participants was 138.1 ± 81.7 days (range 14–364 days). The mean time post-vaccination for all T cell-positive individuals was 147 ± 80.9 days (range 14–364 days). The mean time intervals after vaccination for the three groups are as follows: group A: 52.7 ± 25.4 days (range 14–90), group B: 141.7 ± 25.5 days (range 95–180), and group C: 240.4 ± 41.4 (range 181–364) days.

No significant correlations were established between T cell response and time after vaccination both in the total participants’ group and in the three groups separately (Spearman’s rank correlation test; for total participants: SR = 0.05 *p* = 0.45; for group A: SR = 0.170 *p* = 0.38; for group B: SR = 0.09 *p* = 0.45; for group C: SR = −0.071 *p* = 0.63). Kinetics of T cell responses over time for total participants and for each of the three groups, A, B and C, are exhibited in Figure 2, Figure 3, Figure 4 and Figure 5, which display the curve estimation test for linear regression models.

## 4. Discussion

Over the course of the COVID-19 pandemic, vaccination strategies have been proven more effective against severe disease and mortality and less effective against protection from infection [16]. Taking note of that and relying on the findings from our prior investigation about the duration of cellular immunity [17], we focused herein particularly on fully vaccinated, non-exposed to SARS-CoV-2 infection individuals and their induced cellular and humoral immunity over time.

Our current retrospective study has revealed that T cell immunity following SARS-CoV-2 vaccination persists for at least six months post-vaccination, irrespective of the decline in anti-S antibody levels. Significantly, our research underscores the endurance of cellular immunity for over 8 and up to 12 months post-vaccination, even in the absence of humoral responses. Furthermore, the absence of any significant positive or negative correlations between the T cell response to SARS-CoV-2 and the time post-vaccination in each of the three groups, as well as in the total participant cohort, suggests that cellular immune profiles remained consistently stable throughout the examined study period. An interesting observation in our study is the absence of detectable antibody titers in the individuals of group A within the first three months, contrary to the typical humoral response seen in the general population after vaccination during the initial trimester. While detailed information regarding the specific vaccines administered was not accessible for all participants, the vast majority of them are most likely to have received mRNA vaccines, as this type of vaccines were accessible in Greece during the study period. Despite the exclusion of immunocompromised individuals, it is possible that some participants in group A may have secondary immunodeficiencies not covered by our questionnaire. Additionally, it has been documented [18] that a percentage of healthy individuals in the general population may be poor responders to SARS-CoV-2 vaccines, similar to observations with hepatitis B vaccines, for reasons not clearly elucidated. Moreover, it is possible that, although participants responded to the vaccination, their antibody levels might have declined earlier than the typical three month period.

There are only limited studies reporting long-term cellular and humoral immunity of more than six months after a complete vaccination solely in SARS-CoV-2-naive individuals. In an early study, durable humoral and cellular immune responses were elicited in SARS-CoV-2-unexposed individuals with minimal waning up to eight months after complete vaccination [19]. Cellular immunity was also detectable at eight months after the second dose in SARS-CoV-2-naive individuals, while anti-spike antibody levels declined significantly from 1 to 6 months [20]. A recent large prospective study of UK healthcare workers showed that T and memory B cell responses were maintained six months post primary vaccination, while binding and neutralizing antibodies declined quickly, regardless of the vaccination regime [21]. Similarly, other studies showed that cellular immune responses were present in SARS-CoV-2-naive participants with declining antibody levels six months post complete vaccination [22,23]. Moreover, specific CD4^+^ and CD8^+^ T cells were measurable seven months after a two-dose vaccination, while impaired humoral immunity was reported at four months post-vaccination [24].

Interestingly, in a small percentage (12.5%; 6/48, group C) of our study participants, T cell responses were present up to 10 months after vaccination, highlighting the preservation of cellular immunity for longer than expected. A prospective study enrolling healthcare workers agrees with our results, reporting the preservation of cellular immunity nine months after vaccination in uninfected individuals, especially those who had received an mRNA-based vaccine [25]. Similarly, mRNA vaccines conferred high cellular responses almost eight months post-vaccination, whereas antibody levels had drastically waned [26]. On the other hand, in a multicenter, longitudinal study, robust cellular responses with detectable humoral immunity were documented up to eight months after complete vaccination [27].

On the contrary, other reports reveal waning in both cellular and humoral immunity after a primary vaccination. A recent study, in which a T-SPOT test was also performed, a significant decline in SARS-CoV-2-specific T cells was revealed 6–9 months after the second dose, wherein the median SFCs decreased from 140 (three months post-vaccination) to 48 [28]. A prospective observational study conducted with healthcare workers demonstrated a notable decrease in both humoral and cellular immune responses between 4 and 7 months following complete vaccination, in contrast to the decline observed between 3 and 4 months [29]. Moreover, it was recently demonstrated that more than half of the SARS-CoV-2-naive vaccines had lost their cellular immunity nine months post-vaccination [30]. In young SARS-CoV-2-naive adults particularly, T cell responses were not correlated with peak antibody titers that remained high even six months after vaccination, with a gradual decline [31]. Notably, the vast majority of our study participants that had received an mRNA vaccine presented the highest levels of cellular responses 3–6 months post-vaccination. This finding is in line with the results of a longitudinal study wherein memory CD4^+^ and CD8^+^ T cell responses were high and stable from 3 to 6 months post-vaccination in SARS-CoV-2-naive mRNA vaccines [32]. On the contrary, there are studies demonstrating that cellular immune responses wane six months after full vaccination with a rapid decline between 3 and 6 months post-vaccination [12,33]. A recent investigation revealed the presence of T cell responses lasting up to six months after the second dose, even as antibody levels diminished. The study indicated the highest level of cellular immunity within the first two months post-vaccination, followed by a gradual decline thereafter [34].

The inconsistencies observed in studies investigating the duration of T cell immunity after COVID-19 vaccination can be ascribed to several contributing factors. Variances in study designs, encompassing differences in methodologies and protocols, may impact the interpretation and comparability of results. Additionally, the characteristics of study populations, such as age distribution, health statuses, and varying levels of prior exposure to the virus, play a crucial role in influencing T cell responses. The diverse array of COVID-19 vaccines utilized across studies introduces another layer of complexity, as different vaccines may elicit distinct immune reactions. Individual variability further complicates the landscape, with genetic factors and underlying health conditions contributing to heterogeneous immune responses. Moreover, discrepancies in methodologies for assessing T cell responses, spanning aspects like sample collection, processing, and analysis techniques, can introduce variability in the reported findings. Understanding and addressing these multifaceted factors is essential for developing a comprehensive and unified understanding of T cell immunity longevity in the context of COVID-19 vaccination.

Our study has certain limitations, primarily originating from its observational, single-center, and retrospective nature. In this context, there was a lack of additional information regarding the specific types of vaccines administered to participants, except for the general knowledge that the majority of vaccines given during that period were mRNA-based. Additionally, there was an absence of data regarding participants’ antibody development post-vaccination and the timeline of the vaccination-induced antibody response.

## 5. Conclusions

Our findings underscore the enduring and sustained nature of adaptive cellular immunity in individuals who were SARS-CoV-2-naive, even six months after completing their full vaccination, despite diminished humoral responses. Notably, the highest levels of T cell immunity were observed between 3 and 6 months following the completion of the vaccination regimen and a subset of our cohort maintained elevated T cell responses for 10 months post-vaccination and beyond. The significance of our results lies in the prolonged detectability of cellular immunity for over six months in SARS-CoV-2-naive individuals following primary full vaccination without booster doses. This suggests that additional doses could further enhance T cell protection. Further research is essential to monitor the longevity of SARS-CoV-2 cellular immunity levels after multiple booster vaccine doses. This will optimize immunization schedules and provide a foundation for developing next-generation vaccines against COVID-19 that can elicit robust and enduring cellular immunity rather than relying solely on humoral immunity.

## Figures and Tables

**Figure 1 vaccines-12-00270-f001:**
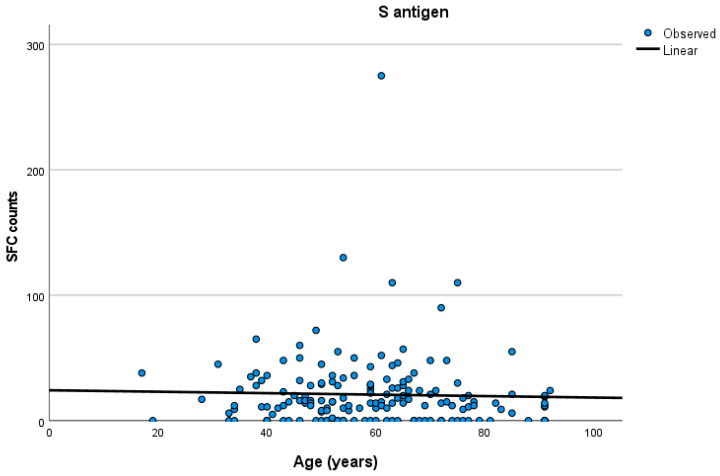
Quantitative T-SPOT analysis of T cell responses to the SARS-CoV-2 S antigen at the respective ages of the total participants. Quantitative T-SPOT results are represented by dots; linear regression model curve estimation for the entire participant cohort.

**Figure 2 vaccines-12-00270-f002:**
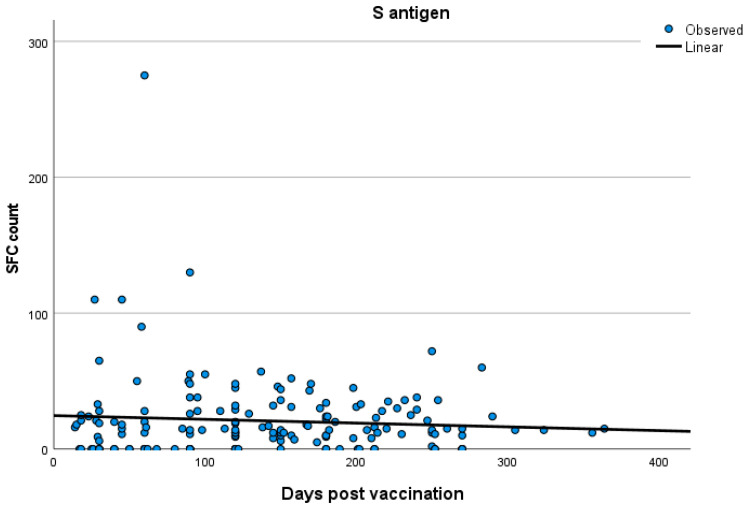
Kinetics (in days) of T cell response with quantitative T-SPOT results represented by dots in figures: linear regression model curve estimation for the entire participant cohort.

**Figure 3 vaccines-12-00270-f003:**
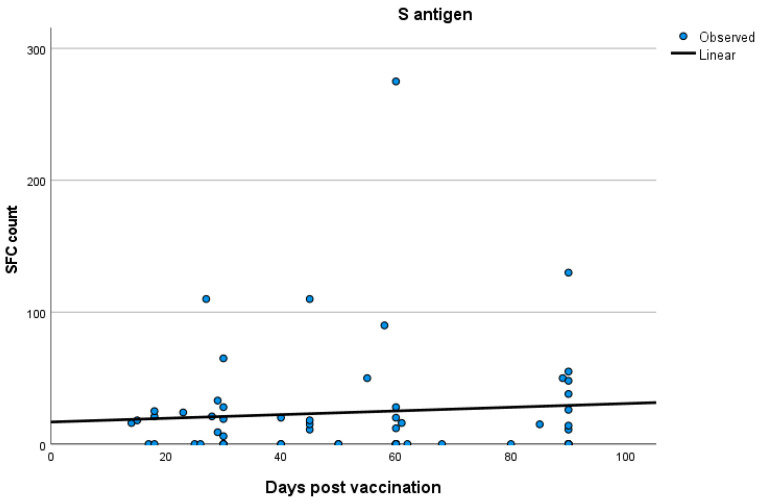
Kinetics (in days) of T cell response with quantitative T-SPOT results represented by dots in figures: linear regression model curve estimation for group A.

**Figure 4 vaccines-12-00270-f004:**
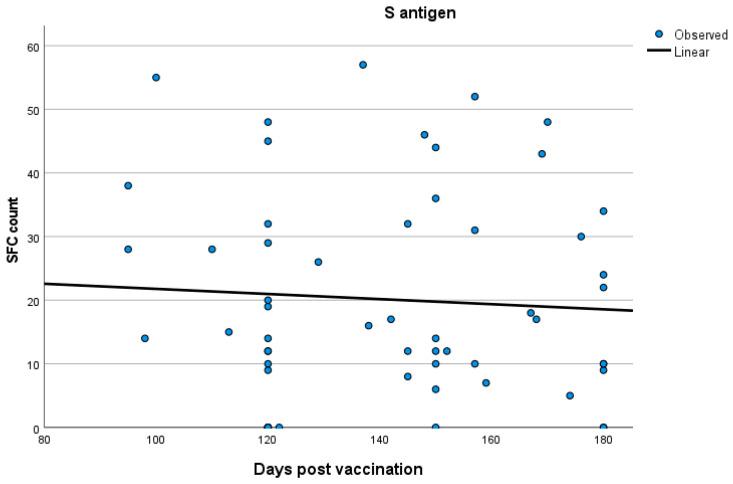
Kinetics (in days) of T cell response with quantitative T-SPOT results represented by dots in figures: linear regression model curve estimation for group B.

**Figure 5 vaccines-12-00270-f005:**
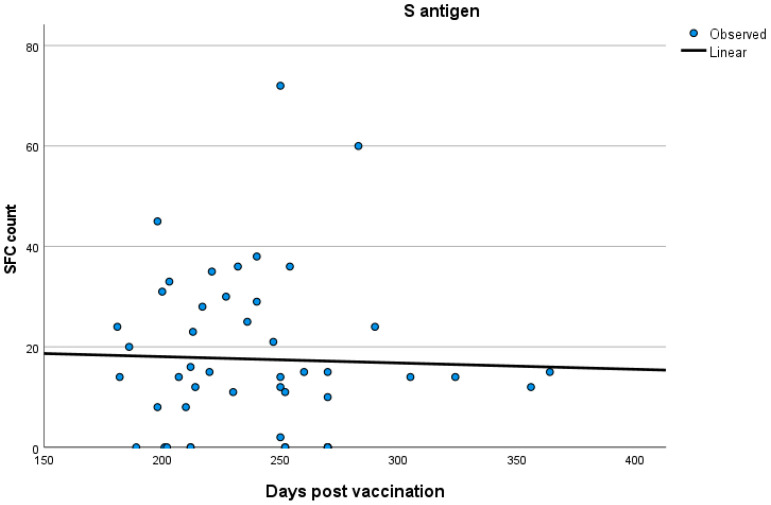
Kinetics (in days) of T cell response with quantitative T-SPOT results represented by dots in figures: linear regression model curve estimation for group C.

**Table 1 vaccines-12-00270-t001:** Participants’ demographic data and comorbidities.

VARIABLES	Group A (N = 60)	Group B (N = 57)	Group C (N = 48)	*p*-Value
Demographic Characteristics				
Age (years ± SD)	58.0 ± 15.2	59.1 ± 17.5	59.58 ± 14.52	NS
Sex (F/M)	36/24 (60/40)	30/27 (52.6/47.4)	25/23 (52.1/47.9)	NS
Comorbidities				
Respiratory disorders, n (%)	8 (13.3)	4 (7)	4 (8.3)	NS
CVD, n (%)	10 (16.6)	4 (7)	3 (6.25)	NS
CNS disorders, n (%)	2 (3.3)	2 (3.5)	0 (0)	NS
Malignant neoplasia, n (%)	2 (3.3)	2 (3.5)	0 (0)	NS
Diabetes mellitus, n (%)	15 (25)	9 (15.7)	8 (16.6)	NS
Hypertension, n (%)	21 (35)	23 (40.3)	18 (37.5)	NS
Lipidemia, n (%)	25 (41.6)	17 (29.8)	15 (31.2)	NS
Obesity, n (%)	25 (41.6)	18 (31.5)	17 (35.4)	NS
Allergies, n (%)	11 (18.3)	8 (14)	5 (10.4)	NS

*p* < 0.05 is considered to be statistically significant; Abbreviations: NS, not significant; F/M, Female/Male; N, number of subjects; SD, standard deviation; CVD, cardiovascular disease, CNS, central nervous system.

## Data Availability

All data from this study are included in this article.

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
