# Peer review of "Prolonged SARS-CoV-2 T Cell Responses in a Vaccinated COVID-19-Naive Population"

_vaccines, 2024, doi:10.3390/vaccines12030270_

Round 1

Reviewer 1 Report

Comments and Suggestions for Authors

This is a highly focused study which only examines primary cellular immune responses in follwing covid vaccination. 

Over 20% of subjects failed to mount significant levels of IFN-g producing T cells. It would be interesting to learn if these subjects were susciptible to more severe covid disease.

Author Response

Point-by-point replies to Reviewer

This is a highly focused study which only examines primary cellular immune responses in following COVID vaccination. 

Over 20% of subjects failed to mount significant levels of IFN-g producing T cells. It would be interesting to learn if these subjects were susciptible to more severe covid disease.

>> Thank you for positive consideration of our submission. Your suggestion is very interesting and we would like to have this information for the individuals who have failed to mount significant levels of T cell immunity. However, this was not possible as our data is limited to information derived from the questionnaires completed by the participants during their examination.

Reviewer 2 Report

Comments and Suggestions for Authors

Here the authors assessed T cell mediated immunity to SARS CoV2 S and N antigen at various times following the primary vaccination series, at 3 time groups.  The goal is to assess T cell immunity when the antibody response has waned.  The findings are that significant T cell responses are present when antibodies have waned.  The manuscript needs some revision.

1. What is meant by fully vaccinated to the primary series. Please define this.

2. Authors state that IgG antibody is negative on day of testing.  Did the subjects initially have any antibody responses to begin with ?

3. The age range is 17-92 years.  This is quite an age range and differences due to age may exist. Please provide a graphic with age vs SFC.

4. At line 186 paragraph, groups are identified as 1, 2, 3.  This should be A, B, C.  

5. At Figure 2, legend should be centered.  Also legends should have some information on Methodology. Also, Fig. 2 legend has duplicated "Error Bars, 95%". Also in Fig. 2 individual data of subjects (dots) should be provided with the Bars and Error Bars.  

6. At line 99- change "was defined" to "defined"

7. At Table1, check Group C 59.58+14.52.

Comments on the Quality of English Language

English is adequate. Only a few typos.

Reviewer 3 Report

Comments and Suggestions for Authors

In this paper, the authors have analyzed T cell responses in vaccinated COVID-19 naive population, with undetectable levels of SARS-CoV-2 IgG. It is an interesting study, but some additional data would be useful to improve the content of this article.

Major comments:

Patients were divided in 3 groups, according to the period between their vaccination and the analysis of T-cell response. It is clear that humoral response can decrease within the months following the vavcination, but usually antibodies are detected within the first trimester after vaccination. How the author explain that in groupe A, they have 60 patients without antibodies againts SARS-CoV-2,whereas it is indiicated that "immunocompromised patients were not included". The criteria used to define "immunodeficiency" should be described.

- The authors explain that T-Spot COVID are coated with S and N antigen, and that patients are classified as "S and/or N reactive". However,in their figures, on the top, they indicate "S antigen".

- In the discussion, they indicate that some studies did not describe prolonged cellular responses. Somme hypothesis should be discussed to explain the dyscrepencies between the studies.

Minor comments:

- 2 decimal places are too many for ages or days

- Figures 4 to 6 could be condensed into one, using different symbols for different patient groups.

Reviewer 4 Report

Comments and Suggestions for Authors

The manuscript describes a cellular response study to COVID-19 vaccination in subjects who had no previous SARS-CoV-2 infection and simultaneously did not develop a post-vaccination humoral response.

Since the manuscript describes a study on a relatively small number of patients, and a lot of duplicate information (e.g., from the manufacturer's instructions) should be removed, the reviewer believes the manuscript should be in the form of a short report. I have several comments:

1.      The abstract is too long. According to the Instructions for Authors, it should consist of a maximum of 250 words. The abstract in the manuscript contains 385 words. Please shorten it.

2.      Introduction: please enter full names first, then abbreviations.

3.      Lines 87-150: please justify the text in the manuscript.

4.      Line 108: please add 2.2 before “Laboratory Method”.

5.      Lines 109-138: if the procedure used was in accordance with the manufacturer's instructions, it should not be described in detail in the text. If there were any deviations, just list them.

6.      Lines 140-150: comment as above.

7.      Lines 159-161: please justify the text in the manuscript.

8.      Since there is no correlation between T-cell response and time after vaccination, has the correlation between cellular response and comorbidities (since the authors list them in Table 1) been studied? Please investigate such a relationship.

9.      As the results shown in Figure 1 are described in the text, I do not see the justification for presenting them additionally in Figure. Please remove it.

Round 2

Reviewer 2 Report

Comments and Suggestions for Authors

The authors have addressed my original concerns.

Reviewer 3 Report

Comments and Suggestions for Authors

Dear authors,

Thank you for your answers. Because of the high frequency of poor response to the vaccination within the first 3 months, it is important to describe the type of vaccination they received. In the ref 18 indicated by the autors (Vaccines (Basel). 2023;11(2):461. doi: 10.3390/vaccines11020461), it was the inactivated vaccines (BBIBP-CorV), which is very unusual. Did your patients received this type of vaccine or ARN-COVID-19 vaccines ?

Author Response

Point-by-point replies to Reviewer #3

Thank you for your answers. Because of the high frequency of poor response to the vaccination within the first 3 months, it is important to describe the type of vaccination they received. In the ref 18 indicated by the autors (Vaccines (Basel). 2023;11(2):461. doi: 10.3390/vaccines11020461), it was the inactivated vaccines (BBIBP-CorV), which is very unusual. Did your patients received this type of vaccine or ARN-COVID-19 vaccines ?

>> Thank you for your constructive comment. In the new version we mention in the Methods section (lines 102-104), that although we lack detailed data regarding the specific vaccines administered to our study participants, it is most likely that the majority of them received mRNA vaccines, as this type of vaccine was administered to the vast majority of the Greek population during the relevant period. We also incorporated this information in the Discussion section (lines 214-217) and replaced Ref. 18 with an article more pertinent to our statement (line 220).

Reviewer 4 Report

Comments and Suggestions for Authors

The manuscript can be published in its present form.

Author Response

Thank you for positive appreciation of our revised submission.

Round 3

Reviewer 3 Report

Comments and Suggestions for Authors

Many thanks for the answer